# Health Risk Assessment of Heavy Metals in Agricultural Soils Based on Multi-Receptor Modeling Combined with Monte Carlo Simulation

**DOI:** 10.3390/toxics12090643

**Published:** 2024-08-31

**Authors:** Yundong Wu, Yan Xia, Li Mu, Wenjie Liu, Qiuying Wang, Tianyan Su, Qiu Yang, Amani Milinga, Yanwei Zhang

**Affiliations:** 1Center for Eco-Environment Restoration Engineering of Hainan Province, School of Ecology and Environment, Hainan University, Haikou 570228, China; wuyundonguuu@163.com (Y.W.); xyamateur@163.com (Y.X.); wangqiuying0409@163.com (Q.W.); sutianyan@163.com (T.S.); yangqiu0903@163.com (Q.Y.); amanistephen@yahoo.com (A.M.); 2Key Laboratory for Environmental Factors Control of Agro-Product Quality Safety (Ministry of Agriculture and Rural Affairs), Tianjin Key Laboratory of Agro-Environment and Safe-Product, Institute of Agro-Environmental Protection, Ministry of Agriculture and Rural Affairs, Tianjin 300191, China

**Keywords:** farmland soil, heavy metals, pollution sources, risk assessment, spatial characteristics

## Abstract

The spatial characteristics, pollution sources, and risks of soil heavy metals were analyzed on Hainan Island. The results showed that the heavily polluted points accounted for 0.56%, and the number of mildly and above polluted points accounted for 15.27%, respectively, which were mainly distributed in the northern part of the study area. The principal component analysis–absolute principal component score–multiple linear regression (APCS-MLR) and the positive matrix factorization (PMF) revealed four sources of heavy metals: agricultural pollution sources for cadmium, (Cd), industrial and mining pollution sources for arsenic, (As), transportation pollution sources for zinc and lead (Zn and Pb), and natural pollution sources for chromium, nickel, and copper (Cr, Ni, and Cu). The human health risk assessment indicated that the average non-carcinogenic risk (HI) for both adults and children was within the safe threshold (<1), whereas Cr and Ni posed a carcinogenic risk (CR) to human health. In addition, the total non-carcinogenic risk (THI) indicated that heavy metals posed a potential non-carcinogenic risk to children, while the total carcinogenic risk (TCR) remained relatively high, mainly in the northern part of the study area. The results of the Monte Carlo simulation showed that the non-carcinogenic risk (HI) for all heavy metals was <1, but the total non-carcinogenic risk index (THI) for children was >1, indicating a potential health risk above the safe threshold. Meanwhile, nearly 100% and 99.94% of the TCR values exceeded 1 × 10^−4^ for children and adults, indicating that Cr and Ni are priority heavy metals for control. The research results provide the necessary scientific basis for the prevention and control of heavy metals in agricultural soils.

## 1. Introduction

With the acceleration of urbanization and industrialization, the overconsumption of fossil fuels, fertilizers, and pesticides, the large amounts of municipal solid waste incinerated and landfilled, and the discharge of industrial effluents and residues, have resulted in the release of more and more heavy metals into soil ecosystems [1,2,3]. Heavy metal contamination in soils is a global concern because of its high toxicity, hidden nature, persistence, bioaccumulation, and widespread presence [4,5]. In China, nearly 16.1% of agricultural soils are contaminated, with heavy metals being the main pollutants. A national geologic survey conducted in the United States revealed that approximately 28% of soil is contaminated with heavy metals [6]. The most widely distributed soil heavy metals pose a threat to human health through three pathways: ingestion, inhalation, and dermal contact [7]. Therefore, there is an urgent need for a comprehensive macro-assessment of the contamination status and ecological risks of agricultural soils to protect agro-ecosystems for sustainable development. This is crucial for human health, particularly in children, who are more prone to the accidental ingestion of heavy metals in the soil [8,9,10].

In recent years, various indices for evaluating soil heavy metals, such as the geo-accumulation index (Igeo), Nemerow index (NIPI), and potential ecological pollution index (RI), have been widely used in practice [11,12]. The qualitative identification of pollution sources and pollution source analysis are important elements in soil heavy metal pollution management. The former uses the geographic information system (GIS) and principal component analysis (PCA) to identify the main pollution sources, while the latter mainly uses receptor models such as positive matrix factorization (PMF), edge analysis (UNMIX), absolute principal component scoring–multiple linear regression (APCS-MLR), and chemical mass balance (CMB) to quantify the pollution sources [13,14]. Among them, the PMF model recommended by the U.S. Environmental Protection Agency (USEPA) has gained great popularity in determining the sources of heavy metals [15], The PMF model utilizes correlation and covariance matrices to simplify the high-dimensional variables by transforming them into multiple composite factors [16]. The model not only ensures non-negative factor distributions and non-negative contributions but also addresses missing and inaccurate values [17]. However, the accuracy of the PMF analysis results can be significantly affected by the sample size and it is difficult to validate the sources. The APCS-MLR model is a receptor model that combines two statistical methods, principal component analysis–factor analysis, and multiple linear regression, to refine the potential sources identified using principal component analysis and quantify their contributions [18]. However, short non-negative constraints are a significant drawback of APCS modelling. No single method can accurately and independently identify the sources of heavy metals in site soils. Therefore, in this study, GIS, APCS-MLR, and PMF models were used to investigate the main sources of heavy metals in soil from agricultural land on Hainan Island [19]. The health risk assessment is a model used to evaluate the potential impact of heavy metals on human health through various pathways such as ingestion, dermal contact, and inhalation [20,21]. Previous studies have primarily relied on heavy metal concentrations and exposure parameters, using models to conduct deterministic health risk assessments [6,22]. However, considering the differences in age, health status, gender, and metabolic rates among individuals, the risks faced by different populations and individuals vary, which may lead to deviations between deterministic assessments and actual risks [23]. Probabilistic risk assessment can provide a more accurate estimation of health risks [24]. Monte Carlo simulation, as a standard probabilistic assessment method, can effectively reduce uncertainty in risk assessments and enhance the precision of the results [25,26].

Hainan Island is the second largest island in southern China, with agricultural cultivation and tourism as its main economic sources. In the context of the construction of a free trade harbor in Hainan, the problem of soil pollution has become increasingly prominent, with soil heavy metal pollution being particularly prominent. However, there are relatively few previous studies on heavy metal residues in agricultural soils on Hainan Island, and there is a lack of systematic descriptive information on the distribution characteristics, spatial distribution, sources, and human health risk evaluation of heavy metals in agricultural soils on Hainan Island [27,28,29]. In addition, few studies have considered the spatial information of sampling sites or combined different methods to analyze heavy metal sources. The main objectives of this study were to (1) comprehensively analyze the level and spatial characteristics of soil contamination in farmlands on Hainan Island using multivariate analysis and geostatistical methods, (2) identify and quantify heavy metal pollution sources using APCS-MLR, the PMF model, and Spearman correlation analysis, (3) evaluate the carcinogenic and non-carcinogenic risks for adults and children using Hakanson’s Potential Ecological Risk (PERA) and Health Risk (HRA) models, and explore the assessment of the spatial distribution characteristics of these risks, and (4) assess the probabilistic carcinogenic and non-carcinogenic health risks of heavy metals to adults and children using Monte Carlo methods. The results of the study can provide important theoretical and practical bases for the identification of soil heavy metal pollution sources, risk assessment, and implementation of comprehensive management strategies in similar rapidly developing economic regions.

## 2. Materials and Methods

### 2.1. Overview of the Study Area

Hainan Island (108°37′–111°03′ E, 18°10′–20°10′ N) is located in the southernmost part of China and covers an area of 33,920 square kilometers (Figure 1). The climate is characteristic of a tropical maritime monsoon climate, with an average annual temperature of 23–26 °C and an average annual precipitation of 1500–2500 mm. Production activities on Hainan Island are predominantly agricultural, with agricultural land accounting for more than 4/5 of the island’s total area [30]. The production of crops, mainly rice, winter melons, vegetables, and fruits is very well developed and therefore also includes the extensive use of pesticides, fertilizers, and other agrochemicals. Agricultural production methods mainly include crop rotation and intercropping, and irrigation relies mainly on water resources such as reservoirs, natural rivers, and rainfall. In addition, industrial activities such as metal mining, related smelters, paper mills, cement factories, etc., contribute significantly to the development of the local economy. The soil is mainly acidic, with a soil pH of 4.2–6.3. The main soil types are red brick soil, red soil, yellow brick soil, coastal sandy soil, coastal salt soil, black lime soil, volcanic ash soil, and rice soil. The plains are interspersed with hills and the terrain is complex, mainly characterized by a high terrain in the middle area and a low terrain around the middle area [31].

### 2.2. Sample Collection and Analysis

In this study, 360 soil samples were collected in March 2023 from agricultural land in the study area. The distribution of soil sample points on agricultural land in Hainan Province is shown in Figure 1. The random distribution of points was carried out with a consideration for equilibrium. Five surface soil samples were collected from a 0 to 20 cm depth at each point using the plum sampling method. Each sampling point consisted of five sub-samples, which were mixed to form a single soil sample, and which was brought back in a polyethylene sampling bag with a mass of ≥1500 g of each original fresh sample; the location of the points was recorded using GPS during the sampling process. All soil samples were air-dried, passed through a 100-mesh sieve, and stored in polyethylene sampling bags.

The soil pH level was measured using a pH meter in a 1:2.5 aqueous suspension. All metals were determined after the sample pretreatment. Cd samples were treated with HF, HNO3, and HClO4 as samples with aqua regia, and Cr, Ni, Cu, Zn, and Pb samples were treated with the powder pressure method. Inductively coupled plasma mass spectrometry (ICP-MS, Agilent Technologies 7700 Series, Santa Clara, CA, USA) was utilized to quantify the concentration of Cd, atomic fluorescence spectrometry (AFS-8220 Beijing Titan Instruments, Beijing, China) was utilized to quantify the concentration of As, and X-ray fluorescence spectrometry (HS-XRF, MERAK-SC Beijing Anke Huisheng Technology, Beijing, China) was utilized to quantify the concentration of Cr, Ni, Cu, Zn, and Pb. Quality control was performed by inserting one standard reference material and one duplicate sample into each 50 samples to calculate the passing rate of accuracy, precision, and relative deviation. The passing rate of accuracy and precision was calculated to be 100%, and the passing rate of relative deviation was 95%, which was in accordance with the requirements of the soil standard DZ/T 0295–2016 [32]. The method detection limits (MDLs) were 2.00, 2.00, 2.00, 2.00, 0.05, 2.00, and 0.003 mg/kg for Cr, Ni, Cu, Zn, As, Pb, and Cd, respectively.

### 2.3. Methods for Evaluating Heavy Metal Pollution of Soils

#### 2.3.1. Pollution Index and Nemerow Composite Pollution Index

The pollution index (PI) and the Nemerow composite pollution index (NIPI) have also been used to investigate the quality of soil samples. The NIPI allows for the assessment of the overall level of soil contamination, taking into account the content of all heavy metals [33].

The individual pollution index (PI) reflects the degree of contamination of individual heavy metal elements and is calculated by the following formula:$$PI=C_{i}/S_{i}$$
where PI is the single pollution index of heavy metal i; Ci is the measured content of heavy metal i in the soil; and Si is the screening value of the pollution risk of heavy metal i in the soil. As shown in Table 1, the pollution level was divided into four levels according to the PI values from 0 to 3: Pi ≤ 1, 1 < Pi ≤ 2, 2 < Pi ≤ 3, and Pi > 3 corresponded to “nonpollution”, “light pollution”, “moderate pollution”, “heavy pollution”, and “heavy pollution”, respectively. The 4 levels were “light pollution”, “moderate pollution”, “heavy pollution”, respectively [34].

The Nemerow pollution index method is a weighted multifactorial environmental quality index that takes into account extreme or significant maxima to highlight the impact of the highest levels of pollutants on soil environmental quality.

Its calculation formula is as follows:$$NIPI=\sqrt{\frac{{\left(C_{i}/S_{i}\right)}_{max}^{2}+{\left(C_{i}/S_{i}\right)}_{ave}^{2}}{2}}$$

${\left(C_{i}/S_{i}\right)}_{max}^{2}$ is the maximum value of the single pollution index of heavy metals in soil and the ${\left(C_{i}/S_{i}\right)}_{ave}^{2}$ is the average value of the single pollution index of soil heavy metals. As shown in Table 1, according to NIPI, the value of 0–3 classifies the pollution level into 5 levels [35].

#### 2.3.2. Index of Geo-Accumulation

The index of geo-accumulation ($I_{geo}$) is given by the following formula:$$I_{geo}=\log_{2}\left[\frac{C_{i}}{\left(1.5B_{i}\right)}\right]$$
where $C_{i}$ is the measured content of heavy metal i in the soil (mg·kg^−1^) and $B_{i}$ is the background value of heavy metal i in soil (mg·kg^−1^). As shown in Table 2, $I_{geo}$ can be divided into 7 levels. The background values adopted by the research were grounded on soil element background values in Hainan [36].

#### 2.3.3. Ecological Risk Assessment Methods for Heavy Metals in Soils

The potential ecological hazard index method was created by Swedish scientist Hakanson. This method combines the environmental ecological effects of heavy metals with toxicology to measure the potential harm of heavy metal pollutants to organisms [37]. The formula is as follows:$$RI=\sum E_{r}^{i}$$
$$E_{r}^{i}=T_{r}^{i}\times C_{f}^{i}=T_{r}^{i}\times C_{i}/C_{n}^{i}$$
where RI is the combined potential ecological hazard index of multiple heavy metals in the soil at a given point; $E_{r}^{i}$ is the potential ecological hazard index of a specific heavy metal in the soil; $T_{r}^{i}$ stands for the toxicity coefficient of heavy metal i; $C_{f}^{i}$ denotes the pollution coefficient of heavy metal i; $C_{i}$ is the measured level of heavy metal i in the soil; and $C_{n}^{i}$ is the standard value for the control of soil pollution risk of heavy metal i in the agricultural land (mg·kg^−1^). The toxicity coefficients for heavy metals Cr, Ni, Cu, Zn, As, Pb, and Cd are 2, 5, 5, 1, 10, 5, and 30, respectively [38]. As shown in Table 3, the potential ecological risk can be categorized into five levels according to the magnitude of the potential ecological hazard index of heavy metals.

### 2.4. Source Analysis of Soil Heavy Metals

#### 2.4.1. Absolute Factor Score Multiple Linear Regression (APCS-MLR) Model

The absolute factor score–multiple linear regression (APCS-MLR) model uses principal component analysis (PCA) to obtain the absolute principal component factor score (APCS) and then uses APCS as an independent variable and the content of heavy metals as a dependent variable to conduct a multiple linear regression analysis to calculate the contribution rates of different pollution sources [39]. The following are the steps to be taken.

Firstly, the data for each heavy metal content should be normalized as follows.
$$Z_{ij}=\left(C_{i_{j}}-{\bar{C}}_{j}\right)/\delta_{j}$$

In the formula, $Z_{ij}$ indicates a standardized value, $C_{i_{j}}$ denotes the content of the element j in the i sample (mg·kg^−1^), and ${\bar{C}}_{j}$ and $\delta_{j}$, respectively, denote the mean content and standard deviation of the element j (mg·kg^−1^).

Secondly, the absolute principal component factor score, APCS, is calculated by introducing a zero (0) concentration factor.
$$z_{j0}=\frac{0-{\bar{C}}_{j}}{\delta_{j}}=\frac{{\bar{C}}_{j}}{\delta_{j}}$$
$$APCS=z_{ik}-\left(z_{0}\right)$$

With APCS as the independent variable and the heavy metal content of each soil as the dependent variable, the multiple linear regression analysis was carried out, and then the regression coefficients were used to calculate the contribution of each pollutant source to the soil heavy metal, and the calculation formula is as follows:$$C_{j}=b_{0j}+\sum_{k=0}^{P}\left(b_{kj}\times APCS\right)$$

In the formula, $C_{j}$ indicates the concentration of heavy metals j (mg·kg^−1^); $b_{0j}$ indicates the constant term of the multiple regression for heavy metals j; the symbol P denotes the number of factors; $b_{kj}$ indicates the regression coefficient of the source k on heavy metals j; and $b_{kj}\times APCS$ indicates the contribution rate of the source P to the concentration of heavy metal $C_{j}$.

#### 2.4.2. Positive Definite Matrix Factorization Model (PMF)

The PMF is a receptor model developed by the United States Environmental Protection Agency, which is a model based on principal component analysis to identify and quantify the sources of pollution. In this study, PMF 5.0 was used to identify and quantify the main sources of heavy metals in agricultural soils. All methods, principles, and applications of this model follow the USEPA PMF 5.0 User Guide [40]. In summary, the PMF decomposed the dataset $x_{ij}$ into two matrices [41,42].

The formula is as follows:$$x_{ij}=\sum_{k=1}^{P}g_{ik}f_{kj}+e_{ij}$$
where $x_{ij}$ is a measurement matrix of heavy metal element j in i number of samples, $g_{ik}$ is a contribution matrix of source factor k for i number of samples, $f_{kj}$ is a source profile of heavy metal element j for the k source factor, and $e_{ij}$ refers to the residual value of heavy metal element j in i number of samples. The minimum value of the objective function Q can be computed by the following formula:$$Q=\sum_{i=1}^{n}\sum_{j=1}^{m}{\left(\frac{e_{ij}}{u_{ij}}\right)}^{2}$$

The uncertainty $u_{ij}$ of the heavy metals in the soil samples is calculated as follows when the content is less than or equal to the corresponding MDL (method detection limit):U=5/6×MDL

When the content is greater than the corresponding MDL, the following calculation is performed:$$U_{ij}=\sqrt{{\left(errorfraction\times concentrations\right)}^{2}+{\left(\begin{array}{c} \frac{MDL}{2}\\ \; \end{array}\right)}^{2}}$$

Concentration data of five heavy metals in 360 soil samples and the related uncertainty data were input to PMF 5.0, and then the number of factors was set to 2, 3, 4 and 5 in independent model runs. Furthermore, the “random start seed number” option was selected, and the number of runs was set at 20. When the number of factors was 4, the Q value was at the minimum value and stable.

### 2.5. Human Health Risk Assessment Model (HHR)

The present study was based on the health risk assessment model proposed by the US EPA (1997) by using equations to estimate the average daily dose (ADD) for different populations (adults and children) for direct ingestion, dermal contact, and oral–nasal inhalation [14,43]. These three routes of exposure were calculated as follows.
$$ADDingest=\frac{Ci\times IngR\times EF\times ED}{BW\times AT}\times 10^{-6}$$
$$ADDdermal=\frac{Ci\times SA\times AF\times ABS\times EF\times ED}{BW\times AT}\times 10^{-6}$$
$$ADDinhalation=\frac{Ci\times APM\times InhR\times EF\times ED}{BW\times AT}\times 10^{-6}$$
where, ADDingest, ADDdermal, and ADDinhalation are the number of heavy metals absorbed by ingestion, skin contact, and inhalation, respectively, and Ci is the measured content of heavy metals in the soil (mg·kg^−1^). The meanings and reference values of other parameters are shown in Table 4.

The carcinogenic risk index (CR) and the non-carcinogenic risk index (HI) were calculated according to the following formulae:$$HQ=\frac{ADD}{RfD}$$
HI=∑HQ
THI=∑HI
CR=∑ADD×SF
TCR=∑CR

In the formulae, HQ is the hazard index of different exposure pathways of heavy metals, RfD is the reference dose corresponding to the exposure pathways of heavy metals, and SF is the carcinogenicity conversion factor of different exposure pathways of the corresponding heavy metals. The specific reference values are shown in Table 5, where the HI value is less than 1, which indicates the non-carcinogenic risk of the population if HI is greater than 1. Where the HI value is less than 1, it means that the level of non-carcinogenic risk is within the acceptable range, and if HI is greater than 1, there is a non-carcinogenic risk in the population. The THI denotes the total non-carcinogenic health risk index for all exposure pathways of multiple heavy metals. The carcinogenic risk is used to characterize the carcinogenicity index of an individual due to exposure to heavy metals over a lifetime, where CR represents the carcinogenic risk of heavy metals, where if 1 × 10^−6^ ≤ CR < 1 × 10^−4^, the population has an acceptable level of cancer risk, and CR ≥ 1 × 10^−4^ indicates an unacceptable level of carcinogenic risk [47]. The TCR indicates the total carcinogenic risk of multiple heavy metals. In this study, Monte Carlo simulation in the Oracle Crystal Ball risk analysis software application was used to characterize the probability distribution of health risks due to heavy metals in agricultural fields in the study area.

### 2.6. Statistics

Microsoft Excel 2019 and Origin 2022 (Origin Lab, Northampton, MA, USA) were used for data processing and mapping. SPSS 26. 0 (IBM, Armonk, NY, USA) software was used to analyze the basic descriptive statistical variables such as the coefficient, standard deviation, maximum and minimum values, and average value of heavy metal content changes in the soil. Arc-GIS 10. 7 (ESRI Inc., Redlands, CA, USA) was used to construct a spatial distribution map of soil heavy metal pollution and analyze relevant spatial data. EPAPMF 5.0 (USEPA, Washington, DC, USA) was used to identify sources of heavy metals and their contribution.

## 3. Results and Discussion

### 3.1. Soil pH and Total Heavy Metal Content

As shown in Table 6, the average value of soil pH in farmlands on Hainan Island was 5.62, ranging from 3.88 to 7.65, with 87.5% of the points having a pH of less than 6.5, indicating that the pH of farmland soils on Hainan Island is mainly acidic. Hainan Island has a tropical typical oceanic climate, abundant rainfall, strong degree of humidity and heat, rapid decomposition of organic matter, high degree of leaching; therefore, the aluminum-rich role is obvious. With the increase in rainfall, the soil cation exchange tends to decrease, and the decrease in salt base saturation also leads to the increase in soil acidity. Many studies have shown that the availability and risk of heavy metal transport in acidic soils is strong [51,52]. The average concentration of heavy metals in soil decreased in the order of Zn (70.57 mg·kg^−1^) > Cr (65.51 mg·kg^−1^) > Pb (36.81 mg·kg^−1^) > Ni (25.42 mg·kg^−1^) > Cu (19.93 mg·kg^−1^) > As (3.37 mg·kg^−1^) > Cd (0.156 mg·kg^−1^). The mean values of heavy metals Cr, Ni, Cu, Zn, As, Pb, and Cd exceeded the background values for soils in Hainan Province, indicating that there is a significant accumulation of these seven heavy metals in the soil. More specifically, compared with the screening value of soil pollution risk screening for agricultural land specified in the Soil Environmental Quality of Soil Pollution Risk Control Standards for Agricultural Land (GB 15618-2018) [53], the contamination degree of the seven heavy metals in the surface soil in descending order was Ni (10.28%) > Cr (9.17%) > Cd (7.78%) > Cu (7.5%) > Pb (1.94%) > Zn (0.83%) = As (0.83%), which is higher than the corresponding risk screening values. This indicates that the agricultural soils in the study area are contaminated with some heavy metals and should be prevented and controlled by appropriate risk prevention and control strategies. The coefficient of variation (CV) reflects the uncertainty and variability of the probability distribution of soil heavy metals. The seven heavy metals in the study area showed a significant spatial variability (CV > 0.36), with coefficients of variation (CV) ranging from 0.5 to 1.66. The largest CV was for As (1.66), suggesting that these seven heavy metals may be locally contaminated and that their accumulation may be significantly affected by regional differences or human activities [16]. Overall, Cr, Ni, Cd, and Cu were the major pollutants in the study area, which is consistent with previous studies [28].

### 3.2. Distribution Characteristics of Soil Heavy Metal Pollution

#### 3.2.1. Levels of Soil Heavy Metal Contamination

According to the screening value of the soil pollution risk for agricultural land specified in the Soil Environmental Quality of Soil Pollution Risk Control Standards for Agricultural Land (GB 15618-2018) [53], the heavy metal pollution status of farmland soil in the study area was evaluated by using the single pollution index method (PI), the Nemerow integrated pollution index method (NIPI), and the ground accumulation index (Igeo). The results showed that the median values of the PI for the seven soil heavy metals ranged from 0 to 1, indicating low levels of contamination (Figure 2a). The highest median PI value was observed for Pb (0.428), followed by Cd (0.41), Zn (0.31), Cu (0.24), Cr (0.24), Ni (0.16), and As (0.06). As shown in Appendix A, seven soil heavy metal elements, including Cr (9.44%), Cu (7.50%), Cd (7.22%), Ni (10.56%), Pb (1.94%), As (0.83%), and Zn (0.83%), exhibited moderate contamination (1 ≤ PI ≤ 3). In particular, Cd (0.56%) showed a considerable level of contamination (PI ≥ 3), suggesting that these seven heavy metals may be significantly affected by regional differences or human activities [16]. Similarly, based on the Igeo values (Figure 2b, Appendix A), the seven soil heavy metals—Cr (8.61%), Ni (3.61%), Cu (7.78%), Zn (3.89%), As (0.56%), Pb (0.83%), and Cd (28.33%), exhibited a moderate level of contamination (1 < Igeo ≤ 2). Meanwhile, seven heavy metal elements, including Cr (10.00%), Ni (6.67%), Cu (10.00%), Zn (38.06%), As (2.22%), Pb (8.89%), and Cd (33.33%), were weakly to moderately contaminated (0 < Igeo ≤ 1). In addition, the relationship between the PI, Igeo, and soil heavy metal elements is shown in Figure 2c. For the PI, the largest contribution of soil heavy metal elements was Cd, followed by Pb, Ni, Cr, Cu, Zn, and As. For the Igeo, the largest contribution was Cd, followed by Ni, Zn, Cr, Cu, Pb, and As. These results indicated that Cd was significantly enriched in the soil and contributed the most to soil pollution in the study area.

The Nemerow integrated pollution index (NIPI) reflects the combined pollution status of seven heavy metals in soil. The number of heavily contaminated points, with an NIPI value greater than 3, accounted for 0.56%, and the total number of mildly and moderately contaminated points accounted for 15.27%, indicating that the agricultural soils in the study area were contaminated by heavy metals to a certain extent. Combined with the Nemerow spatial interpolation in Figure 2d, the spatial distribution of soil heavy metals was mainly concentrated in the northern part of the study area. This may be attributable to the high geochemical background or severe soil pollution in the volcanic area [28,54]. Therefore, the sources of soil heavy metals and the associated health risk distribution should be explored step by step.

#### 3.2.2. Characteristics of the Spatial Distribution of Soil Heavy Metals

The spatial distribution of soil heavy metals provides information on the hotspots and extent of soil pollution in the study area. The spatial distribution of Cr, Ni, Cu, Zn, As, Pb, and Cd in the soil is shown in Figure 3. The spatial distributions of Cr, Ni, and Cu are similar, with their concentrations concentrated in the northern part of the region. However, the northern part of the study area belongs to the volcanic region, which is one of the most important areas of distribution of Cenozoic volcanic rocks. Cr, Ni, and Cu are concentrated in this region due to the background of Cenozoic basal volcanic and Mesozoic mesophilic volcanic rock areas with higher values, thus promoting the accumulation of Cr, Ni, and Cu. This is consistent with the findings of previous studies [28]. The distribution of Zn and Pb in the study area is different from that of Cr, Ni, and Cu, which are mainly evenly distributed in the study area. Studies have shown that Zn and Pb have the same sources and pathways [24]. Since Zn and Pb are marker elements for transportation activities, most of the soil sample collection sites were located in agricultural fields on both sides (>20 m) of the transportation roads. With the wear and tear of automobile exhaust emissions, engine tires, etc., elements such as Zn and Pb are enriched in the soil [16,28]. Therefore, Zn and Pb elements may be affected by traffic factors. The distribution of As and Cd in the study area is characterized by a significant point source distribution, which is highly influenced by anthropogenic disturbances. The results of the study show that the extensive use of chemical fertilizers, pesticides, and herbicides in agricultural activities, as well as industrial and mining wastewater discharges from industrial and mining activities, may contribute to the enrichment of As and Cd in the soils of the study area [55].

#### 3.2.3. Assessment of Potential Ecological Risk of Soil

In the single ecological risk ($E_{r}^{i}$) of soil heavy metals, the results showed (Appendix A) that the single potential ecological hazard indices of Cr, Ni, Cu, Zn, As, Pb, and Cd were less than 40, indicating that the ecological risks of Cr, Ni, Cu, Zn, As, Pb and Cd in the agricultural soils of Hainan Island were low and at the level of a slight risk. The average value of the soil ecological potential hazard index is 7.32, ranging from 1.55 to 32.91. Among them, 100% of the points with a slight ecological risk and below are at a slight pollution level. Combined with the spatial interpolation map analysis, as shown in Appendix A, the ecological risk (RI) of heavy metals in agricultural soils on Hainan Island is generally at the level of slight pollution; compared with other places, its high-value area is mainly distributed in the northern part of the study area, which is similar to the spatial distribution of the NIPI. Therefore, focus should be placed on the sources of soil heavy metals and the associated health risk distribution in the northern part of the study area.

### 3.3. Soil Heavy Metal Source Analysis

#### 3.3.1. Multivariate Statistical Analysis

Correlation analysis can reveal the relationship between soil heavy metals and provide a basis for determining their sources [56]. The results of the Spearman correlation analysis are shown in Figure 4b. There was a significant positive correlation (*p* < 0.01, R > 0.5) between Cr and Ni, Cr and Cu, Cr and Zn, Ni and Cu, Zn and Ni, and Cu and Zn. It is tentatively inferred that Cr, Ni, Cu, and Zn in the soil may have the same source. There was also a significant positive correlation (*p* < 0.01, R > 0.5) between Zn and Pb, indicating that these two heavy metals may also have similar sources. There was no significant correlation (R < 0.5) between As and Cd, suggesting that they may have different sources. In general, a strong positive correlation between soil heavy metals indicates similar sources, while a weak correlation indicates different sources. In addition, the results of the hierarchical cluster analysis are shown in Figure 4a. Four clusters, (1) Cr, Ni, Cu, and Zn, (2) As, (3) Pb, and (4) Cd, can be clearly observed in the dendrogram of soil heavy metals, which can be confirmed by the results of the correlation analysis and principal component analysis.

A PCA was further applied to explore the relationship between soil heavy metals, and Kaiser–Meyer–Olkin (KMO) and Bartlett’s tests were performed to check the appropriateness of the principal component analysis before conducting the principal component analysis. The KMO value of 0.71 and chi-square value of 1183.6 (*p* < 0.01) indicated that the data were suitable for the PCA. The results showed that four principal components (PC1, PC2, PC3, PC4) were extracted, explaining about 87% of the total variance (Appendix A). PC1 was controlled by Cr, Ni, Cu, and Zn and explained 42.6% of the total variance. PC2 was controlled by As and explained 23.4% of the total variance. PC3 was controlled by Pb and explained 12.3% of the total variance. PC4 was controlled by Cd and explained 9.6% of the total variance. These results are consistent with the results of the correlation analysis and hierarchical cluster analysis.

#### 3.3.2. Comparison of Different Receptor Models

The PMF and APCS-MLR models were used to obtain the contribution of different sources of pollution to soil heavy metals, and four factors were derived from these two models (Figure 5a,b). The PMF model determined that the S/N ratios of all the soil heavy metals were greater than 2, indicating a stronger category. The four factors in the APCS-MLR model covered 87.9% of the variance of the data. In the receptor model, R2 was high for all soil heavy metals, ranging from 0.55 to 0.99 (Appendix A), indicating that both receptor models showed satisfactory results. To further understand the relationship between the factors derived from these two models, a Spearman correlation analysis was performed, and the results are listed in Table 7. The correlation factors of the APCS-MLR model were significantly correlated with those of the PMF model (*p* < 0.01), with correlation coefficients greater than 0.68. These results indicated that the two receptor models could be validated against each other, and therefore, the average contribution of each factor to the soil heavy metals was calculated (Figure 5c). Factor 1 contributed 78.57%, 87.19%, and 68.73% to Cr, Ni, and Cu, respectively, and to some extent Zn (28.10%). Factor 2 contributed 61.79% and 84.03% to Zn and Pb, respectively, and to some extent to Cu (25.69%). Factor 3 was the main contributor to Cd (74.20%), followed by Pb (10.64%) and As (9.45%). Factor 4 contributed (71.04%) to As.

#### 3.3.3. Explanation of the Origin of Each Factor

Factor 1 was characterized by Cr, Ni, and Cu, followed by Zn (Figure 5c). Previous studies have shown that the accumulation of Cr, Ni, Cu, and Zn in soil may be caused by soil matrices and industrial and agricultural activities [57,58]. Martin et al. (2006) and Hanesch et al. (2001) found that soil parent material is the main source of Cr and Ni [59,60]. In addition, studies have shown that soil parent material is a major influencing factor on the content and distribution of Cr, Ni, Cu, and Zn in the study area [28,54]. The spatial distribution heat maps of Cr, Ni, Cu, and Zn contents generated by the inverse distance-weighted interpolation method were mainly distributed in the northern areas of the study area, which are located in the Cenozoic basaltic and Mesozoic mesophilic volcanic rocks of the study area. The basaltic rocks and associated soils are characterized by high Ni, Cr, and Zn contents [61]. The basal volcanic rocks also have a significantly higher Cu content than the other parent materials [62]. However, studies have shown that the accumulation of Cr, Ni, Cu, and Zn in soils is related to industrial activities [63,64]. Cr and Cu primarily originate from solid waste, industrial wastewater, and the sludge produced by industrial production, and Ni and Zn mainly originate from activities such as electroplating and smelting [57]. However, the northern area of the study region is characterized by a high background value of soil, indicating a significant impact from natural factors. Therefore, factor 1 can be defined as the natural source of the soil parent material.

Factor 2 has the highest contribution of Pb and Zn, followed by Cu (Figure 5c). Since Zn and Pb are marker elements for transportation activities, most of the soil sample collection sites were located in agricultural soils on both sides of transportation roads. With vehicle exhaust emissions, engine and tire wear, elements such as Zn and Pb are enriched in the soil [16,27,28]. According to the analysis of the spatial distribution map of the Pb and Zn content generated by the inverse distance-weighted interpolation method, the spatial distribution of soil heavy metals Pb and Zn is extensive and uniform, indicating that they are greatly influenced by transportation factors. It has been suggested that brake pads, gasoline or fuel leakage, and the wear and tear of engine parts are the main contributors to the heavy metal Cu [7,65]. Additionally, the study area serves as a free trade port and a tourist destination; hence, there are a large number of roads around the farmland. The dense traffic results in the emission of vehicle exhaust, and heavy metal elements such as Zn, Pb, and Cu from car engines and tire wear are released into the soil environment. Therefore, factor 2 is considered as a source of transportation pollution.

Factor 3 was mainly loaded with Cd, followed by Pb and As (Figure 5c). Based on the PI and Igeo values, the agricultural soils examined in this study indicated that the heavy metal Cd in the soil was heavily influenced by anthropogenic activities (Figure 2a,b). Previous studies have reported that the accumulation of Cd, Pb, and As in agricultural soils may be related to agricultural activities such as the application of fertilizers, organic manures, and pesticides. In particular, phosphorus fertilizers have high Cd concentrations, ranging from 0.1 to 170 mg/kg [66]; therefore, the large-scale use of phosphorus fertilizers and pesticides may introduce more Cd, Pb, and As into agricultural soils [67]. Zhao et al. (2018) estimated that the application of fertilizers and manure introduces about 3–4% of Cd and Pb content into the soil annually [68]. Pesticides or herbicides used in agricultural production also contain large amounts of inorganic As compounds such as calcium arsenate, lead arsenate, and sodium arsenate [27,69]. According to our field survey, phosphoric acid fertilizers, compound fertilizers, and organic fertilizers were the major types of fertilizers used to increase the crop yield in the study area. Therefore, the composition of factor 3 was considered to be a result of agricultural activity. Similar results have been reported in other studies [66,67], in which it was observed that the long-term application of fertilizers leads to the accumulation of Cd in agricultural soils.

Factor 4 was mainly dominated by the presence of As (Figure 5c). According to the CV and spatial distribution maps shown, the As content in agricultural soils is mainly localized, indicating that human activities are the main source of As. Usually, As is a prominent element in industrial emissions due to smelting and metal processing activities in mining areas [70]. In fact, according to the Record of Land and Mineral Resources of Hainan Province and our field sampling surveys, there are abandoned mining areas in the areas around the hotspots of the high spatial distribution of As. These As-related industrial and mining activities emit large amounts of As-containing soot and wastewater, which leads to As accumulation in the soil. Therefore, factor 4 is characterized by As-related industrial, mining, smelting, and disposal activities.

### 3.4. Human Health Risk Assessment

The levels of ADD, HI, and CR were calculated based on the ingestion, dermal contact, and inhalation of heavy metal elements in the collected soil samples to assess the health risks to humans. As shown in Appendix A, the average ADD of heavy metals ingested by children and adults is much higher than that of dermal contact and inhalation. Thus, ingestion is the primary route of human exposure to heavy metals, which is a conclusion supported by previous studies [6,71]. However, the ADD intake pathway is higher in children than in adults, potentially posing a greater health risk to children. This suggests that children, due to their behaviors, physiological characteristics, and the duration of exposure, are more susceptible to toxic substances than adults. Consequently, the risk of ADD in children is higher than in adults.

In addition, this study used Monte Carlo simulation to assess the uncertainty of the health risk of heavy metals by considering the parameters and their distributions that influence the health risk assessment model (Figure 6, Figure 7 and Figure 8). The results showed that the cumulative non-carcinogenic risk was greater in children compared to adults. The mean non-carcinogenic risk (HI) for all heavy metals was less than 1, indicating no potential non-carcinogenic risk (Figure 7b–h). The probability distribution showed that the mean total noncancer risk (THI) for adults was <1, while children (44.51%) had a total noncancer risk > 1 (Figure 7a), which was mainly distributed in the northern part of the study area (Figure 6c,d). In addition, based on the probabilistic HRA model in this study, it was found that the 95th percentile THI value for children exceeded the threshold value of 1 (1.07) (Table 8), indicating that children had some total noncancer risk. According to the probability distribution shown in Figure 8a–f, the carcinogenicity of each of the five heavy metals varies. The mean TCR values for children and adults were $1.22\times 10^{-3}$ and $1.87\times 10^{-4}$, respectively, which exceeded the acceptable limit of $1\times 10^{-6}$. Meanwhile, nearly 100% and 99.94% of TCR values exceeded $1\times 10^{-4}$ in children and adults, respectively (Figure 8a), which were mainly distributed in the northern part of the study area (Figure 6a,b), suggesting that the total carcinogenic risk could not be ignored. According to Figure 8b–f, Cr and Ni were confirmed as the main factors of carcinogenic risk by comparing the maximum mean CR values. In all populations, the mean CR values of Cr and Ni exceeded the acceptable thresholds of $1\times 10^{-6}$; in particular, 100% of the CR values of Cr and Ni in children exceeded the risk thresholds of $1\times 10^{-4}$, indicating a high carcinogenic risk. In addition, the CR values for As and lead should not be ignored. Therefore, in assessing the risk of agricultural soils in the study area, close attention should be paid to the exposure of children to heavy metals, and it is recommended that priority be given to soil management and risk control measures in the northern and southwestern parts of the study area, especially for Cr and Ni. This is consistent with previous research findings, which suggest that children may be more susceptible to environmental heavy metal threats due to their unique physiological structure and dietary habits, necessitating enhanced precaution against the health risks posed by Cr, Ni, As, and Pb to the human body. The overall analysis indicates that heavy metals in soil pose a significant health risk to humans, with both non-carcinogenic and carcinogenic risks being higher for children than for adults. This suggests that children, due to their smaller body weight and ongoing physical development, are more likely to absorb heavy metals from soil and crops, thereby having a higher cancer risk and non-carcinogenic risk than adults. It is necessary to implement pollution control measures for the heavy metals Cr, Ni, As, and Pb in farmlands with high geological background concentrations [28].

## 4. Conclusions

This study investigated the current status, sources, and health risks of Cr, Ni, Cu, Zn, As, Pb, and Cd contamination in agricultural soils on Hainan Island. The mean concentrations of Cr, Ni, Cu, Zn, As, Pb, and Cd in agricultural soils exceeded the corresponding background values. The calculated PI and Igeo values indicated that the heavy metals polluted the agricultural soils to varying degrees, with Cd being particularly affected by human activities. The APCS-MLR and PMF models revealed four major sources of heavy metals in farmland soils in Hainan Island, including agricultural pollution sources, industrial and mining pollution sources, transportation pollution sources, and natural sources of the soil matrix. The As content in the soil mainly originates from industrial and mining activities, the Cd content mainly originates from agricultural activities, the lead and Zn content mainly originates from transportation activities, and the Cr, Ni, and Cu content mainly originates from soil matrices. Using the probabilistic risk estimation method of Monte Carlo simulation to assess health risks, the potential health risks of soil heavy metals are greater for children compared to adults. The mean non-carcinogenic risk (HI) values for both adults and children were within the threshold of safety (<1), whereas the total non-carcinogenic risk (THI) indicated that heavy metals pose a potential total non-carcinogenic risk to children. The carcinogenic risk (CR) of Cr and Ni poses a carcinogenic risk to both adults and children in the study area. The total carcinogenic risk (TCR) was higher, mainly in the northern part of the study area. In conclusion, soil heavy metal contamination in agricultural soils and its associated health risks are of great concern and more attention should be given to activities related to Cr and Ni production as a priority source of contamination in the region.

## Figures and Tables

**Figure 1 toxics-12-00643-f001:**
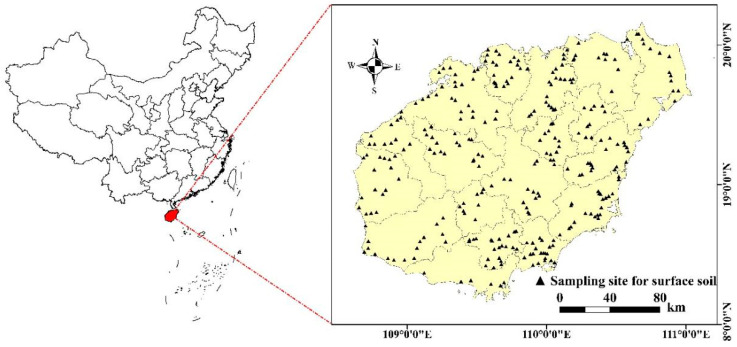
Distribution of soil sampling points on agricultural land in Hainan Province.

**Figure 2 toxics-12-00643-f002:**
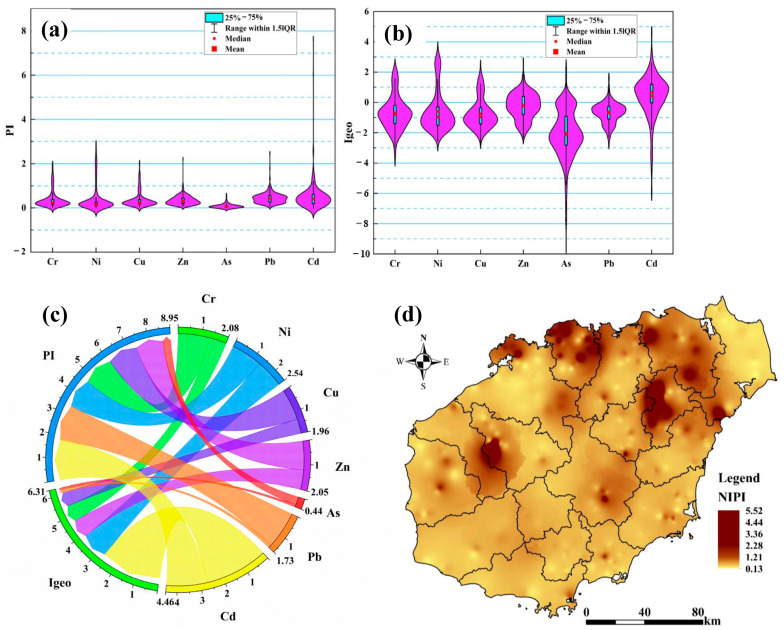
Violin box plot of soil heavy metals in the study area: (**a**) PF, (**b**) Igeo, (**c**) The relationship between soil heavy metals and environmental indices, and (**d**) (NIPI).

**Figure 3 toxics-12-00643-f003:**
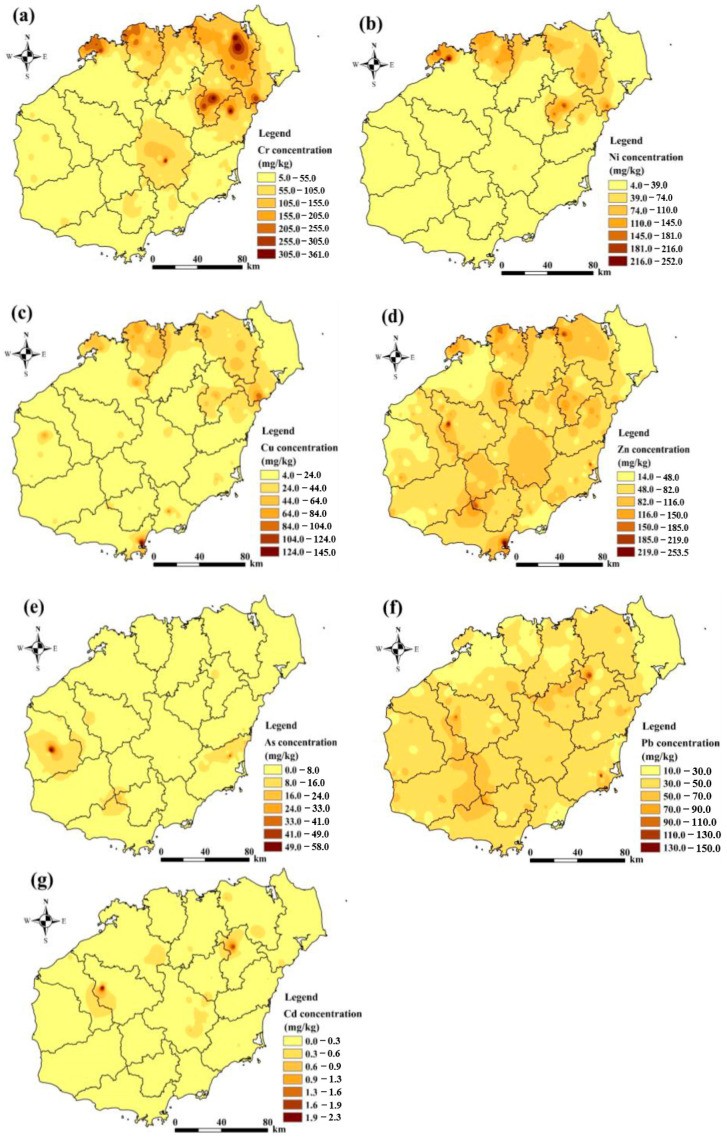
Spatial distribution of seven heavy metals: Cr (**a**), Ni (**b**), Cu (**c**), Zn (**d**), As (**e**), Pb (**f**), and Cd (**g**).

**Figure 4 toxics-12-00643-f004:**
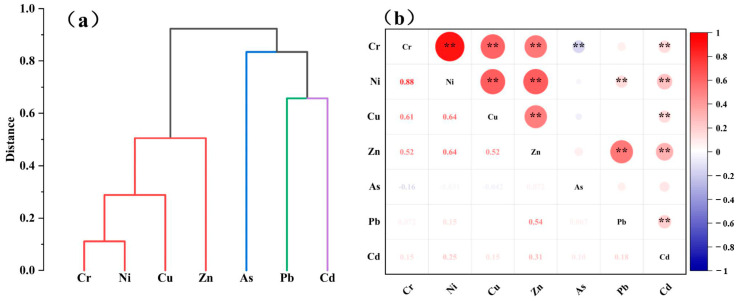
(**a**) Hierarchical clustering analysis and (**b**) soil heavy metal correlation coefficient (** represents a significant correlation between heavy metals; red represents positive correlation, blue represents negative correlation; the darker the color of the circle, the larger the circle, and the stronger the correlation).

**Figure 5 toxics-12-00643-f005:**
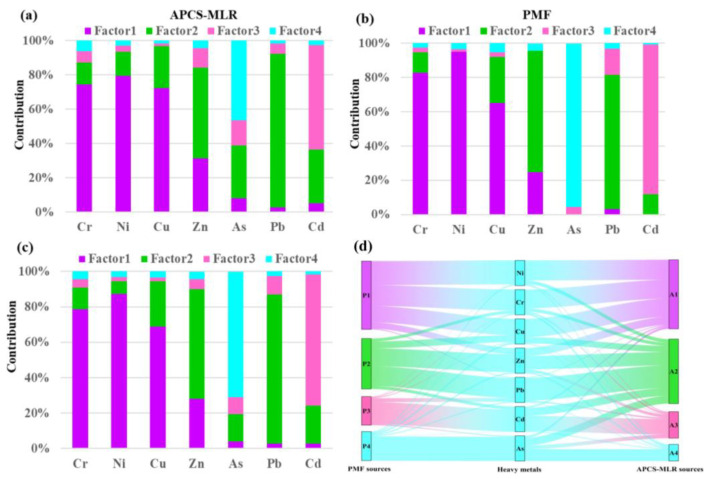
Source contributions (**a**,**b**,**d**) and average factor contributions (**c**) for each factor derived from PMF and APCS/MLR.

**Figure 6 toxics-12-00643-f006:**
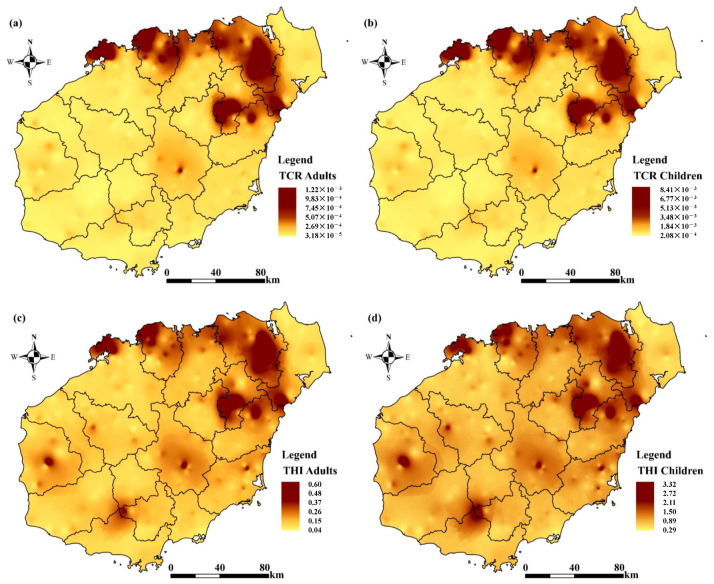
Spatial distribution of total carcinogenic and non-carcinogenic risks in adults and children ((**a**,**b**) spatial distribution of total carcinogenic risk (TCR), and (**c**,**d**) spatial distribution of total non-carcinogenic risk (THI)).

**Figure 7 toxics-12-00643-f007:**
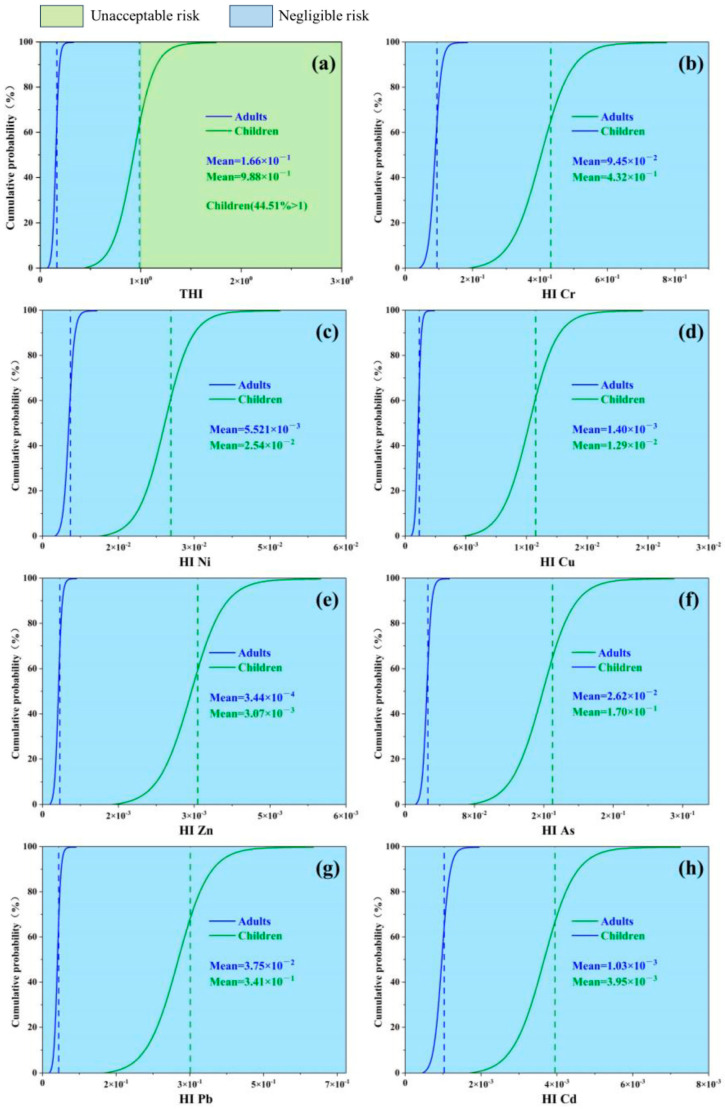
Non-carcinogenic probabilistic health risk assessment for each heavy metal and all heavy metals ((**a**) total non-carcinogenic risk (THI) and non-carcinogenic risk (HI) for (**b**) Cr, (**c**) Ni, (**d**) Cu, (**e**) Zn, (**f**) As, (**g**) Pb, and (**h**) Cd. Blue and green vertical dashed lines indicate mean values for adults and children, respectively).

**Figure 8 toxics-12-00643-f008:**
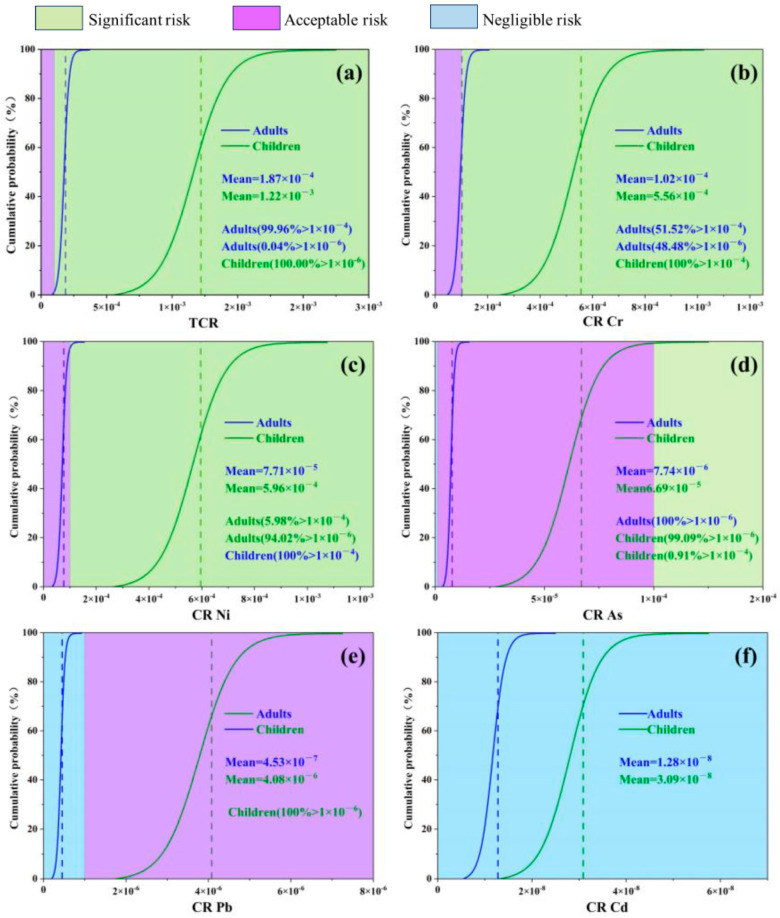
Carcinogenic probabilistic health risk assessment for each heavy metal and all heavy metals ((**a**) total carcinogenic risk (TCR) and carcinogenic risk (CR) for (**b**) Cr, (**c**) Ni, (**d**) As, (**e**) Pb, and (**f**) Cd. Blue and green vertical dashed lines indicate mean values for adults and children, respectively).

**Table 1 toxics-12-00643-t001:** Individual (PI) and composite pollution indices (NIPI).

Classification Level		Value
Individual pollution index (PI)		
Class 1	Non-pollution	PI < 1
Class 2	Moderate pollution	1 ≤ PI < 2
Class 3	Strong pollution	2 ≤ PI < 3
Class 4	Extremely strong pollution	PI ≥ 3
Composite pollution index (INPI)		
Class 1	safety	INPI ≤ 0.7
Class 2	Alert Limit	0.7 < INPI ≤ 1.0
Class 3	Lightly contaminated	1 < INPI ≤ 2
Class 4	Medium Pollution	2 < INPI ≤ 3
Class 5	Heavily polluted	3 < INPI

**Table 2 toxics-12-00643-t002:** Land cumulative pollution index.

Land Cumulative Pollution Index ($I_{geo}$)		
Class 1	Not to weakly contaminated	$I_{geo}$ ≤ 0
Class 2	Weakly to moderately contaminated	0 < $I_{geo}$ ≤ 1
Class 3	Moderately contaminated	1 < $I_{geo}$ ≤ 2
Class 4	Moderately to strongly contaminated	2 < $I_{geo}$ ≤ 3
Class 5	Strongly contaminated	3 < $I_{geo}$ ≤ 4
Class 6	Strongly to extremely contaminated	4 < $I_{geo}$ ≤ 5
Class 7	Extremely contaminated	5 < $I_{geo}$

**Table 3 toxics-12-00643-t003:** Individual and combined potential ecological hazard indices.

Classification Level		Value
Index of single-factor ecological risk ($E_{r}^{i}$)		
Class 1	Low risk	$E_{r}^{i}$ < 40
Class 2	Moderate risk	40 ≤ $E_{r}^{i}$ < 80
Class 3	Considerable risk	80 ≤ $E_{r}^{i}$ < 160
Class 4	High risk	160 ≤ $E_{r}^{i}$ < 320
Class 5	Extremely high risk	$E_{r}^{i}$ ≥ 320
Index of comprehensive ecological risk (RI)		
Class 1	Low risk	RI < 150
Class 2	Moderate risk	150 ≤ RI < 300
Class 3	Considerable risk	300 ≤ RI < 600
Class 4	High risk	600 ≤ RI < 1200
Class 5	Extremely high risk	RI ≥ 1200

**Table 4 toxics-12-00643-t004:** Heavy metal exposure indicators [44,45,46].

Indicator	Meaning	Unit	Adult	Child
lngR	Soil intake rate	mg·d^−1^	100	200
EF	Frequency of exposure	d·year	350	350
ED	Years of exposure	year	24	6
BW	Average body weight	kg	56.8	15.9
AT	Average exposure time	d	24 × 365	6 × 365
SA	Exposed skin surface area	cm^2^	5938	2493
AF	Skin Adhesion Factor	mg·(cm^2^·d)^−1^	0.07	0.2
ABS	Skin absorption factor	-	0.001	0.001
APM	Particulate Volume per Unit Volume	mg·m^−3^	0.0651	0.0651
lnhR	Daily air intake	m^3^·d^−1^	14.5	7.5

**Table 5 toxics-12-00643-t005:** Reference dose (RfD) and slope factor (SF) of soil PTEs in different pathways [48,49,50].

Heavy Metal	RfD (mg/kg/day)			SF (per mg/kg/day)		
	Ingest	Dermal	Inhalation	Ingest	Dermal	Inhalation
Cr	3.0 × 10^−3^	3.0 × 10^−5^	2.86 × 10^−5^	5.01 × 10^−1^	2.0 × 10^1^	4.2 × 10^1^
Ni	2.0 × 10^−2^	5.4 × 10^−3^	9.0 × 10^−5^	1.7	4.25 × 10^1^	8.4 × 10^−1^
Cu	4.0 × 10^−2^	1.2 × 10^−2^	-	-	-	-
Zn	3.0 × 10^−1^	6.0 × 10^−2^	-	-	-	-
As	3.0 × 10^−4^	1.23 × 10^−4^	4.29 × 10^−6^	1.5	1.5	1.51 × 10^1^
Pb	1.4 × 10^−3^	5.24 × 10^−4^	-	8.5 × 10^−3^	-	4.2 × 10^−2^
Cd	1.0 × 10^−3^	2.5 × 10^−5^	2.86 × 10^−6^	-	-	6.3

**Table 6 toxics-12-00643-t006:** Statistical characteristics of soil pH (Unitless) and heavy metal content (mg kg^−1^) in the study area.

Elemental	pH	Cr	Ni	Cu	Zn	As	Pb	Cd
Min	3.88	5.22	3.65	3.92	13.84	0	7.57	0
Max	7.65	363.45	253.87	146.64	446.75	58.66	173.82	2.30
Mean	5.62	65.51	25.42	19.93	70.57	3.37	36.81	0.156
Standard deviation	0.71	62.27	38.77	18.85	44.04	5.67	18.36	0.186
Coefficient of variation		1.02	1.49	0.93	0.63	1.66	0.50	1.18
Hainan soil background		27.50	7.24	6.10	44.4	1.34	24.4	0.04
Soil background in China		61.00	27.00	23.00	74.00	11.00	26.00	0.097

**Table 7 toxics-12-00643-t007:** Correlations coefficients among the factors derived from two receptor models.

		APCS-MLR				PMF			
		F1	F2	F3	F4	F1	F2	F3	F4
APCS-MLR	F1	1							
	F2	−0.116 *	1						
	F3	0.139 **	0.071	1					
	F4	−0.301 **	−0.039	−0.026	1				
PMF	F1	0.848 **	−0.054	0.311 **	−0.266 **	1			
	F2	0.048	0.818 **	−0.166 **	−0.053	−0.012	1		
	F3	−0.150 **	−0.146 **	0.793 **	0.109 *	−0.045	−0.330 **	1	
	F4	0.153 **	−0.035	0.018	0.680 **	0.033	0.058	0	1

** At the 0.01 level. * At the 0.05 level.

**Table 8 toxics-12-00643-t008:** Concentration-specific health risk of soil PTEs for adults and children.

Risk	Soil PTEs		5%	Mean	Median	95%
HI	Cr	Adults	7.97 × 10^−2^	9.36 × 10^−2^	6.22 × 10^−2^	1.04 × 10^−1^
		Children	3.65 × 10^−1^	4.29 × 10^−1^	2.85 × 10^−1^	4.75 × 10^−1^
	Ni	Adults	4.13 × 10^−3^	5.47 × 10^−3^	2.37 × 10^−3^	6.33 × 10^−3^
		Children	1.90 × 10^−2^	2.52 × 10^−2^	1.09 × 10^−2^	2.92 × 10^−2^
	Cu	Adults	6.15 × 10^−4^	7.09 × 10^−4^	4.67 × 10^−4^	7.78 × 10^−4^
		Children	5.58 × 10^−3^	6.43 × 10^−3^	4.24 × 10^−3^	7.06 × 10^−3^
	Zn	Adults	3.22 × 10^−4^	3.41 × 10^−4^	2.98 × 10^−4^	3.63 × 10^−4^
		Children	2.89 × 10^−3^	3.05 × 10^−3^	2.67 × 10^−3^	3.25 × 10^−3^
	As	Adults	1.98 × 10^−2^	2.60 × 10^−2^	1.71 × 10^−2^	3.05 × 10^−2^
		Children	1.29 × 10^−1^	1.69 × 10^−1^	1.11 × 10^−1^	1.99 × 10^−1^
	Pb	Adults	3.56 × 10^−2^	3.71 × 10^−2^	3.47 × 10^−2^	3.90 × 10^−2^
		Children	3.25 × 10^−1^	3.39 × 10^−1^	3.17 × 10^−1^	3.56 × 10^−1^
	Cd	Adults	8.82 × 10^−4^	1.02 × 10^−3^	8.61 × 10^−4^	1.14 × 10^−3^
		Children	3.40 × 10^−3^	3.93 × 10^−3^	3.32 × 10^−3^	4.42 × 10^−3^
THI	Total	Adults	1.41 × 10^−1^	1.64 × 10^−1^	1.18 × 10^−1^	1.82 × 10^−1^
		Children	8.50 × 10^−1^	9.76 × 10^−1^	7.34 × 10^−1^	1.07 × 10^0^
CR	Cr	Adults	8.60 × 10^−5^	1.01 × 10^−4^	6.71 × 10^−5^	1.12 × 10^−4^
		Children	4.70 × 10^−4^	5.52 × 10^−4^	3.67 × 10^−4^	6.11 × 10^−4^
	Ni	Adults	5.76 × 10^−5^	7.63 × 10^−5^	3.31 × 10^−5^	8.84 × 10^−5^
		Children	4.47 × 10^−4^	5.92 × 10^−4^	2.57 × 10^−4^	6.85 × 10^−4^
	As	Adults	5.84 × 10^−6^	7.66 × 10^−6^	5.05 × 10^−6^	9.00 × 10^−6^
		Children	5.06 × 10^−5^	6.64 × 10^−5^	4.37 × 10^−5^	7.80 × 10^−5^
	Pb	Adults	4.30 × 10^−7^	4.49 × 10^−7^	4.19 × 10^−7^	4.72 × 10^−7^
		Children	3.88 × 10^−6^	4.05 × 10^−6^	3.79 × 10^−6^	4.26 × 10^−6^
	Cd	Adults	1.10 × 10^−8^	1.27 × 10^−8^	1.08 × 10^−8^	1.43 × 10^−8^
		Children	2.66 × 10^−8^	3.07 × 10^−8^	2.60 × 10^−8^	3.45 × 10^−8^
TCR	Total	Adults	1.50 × 10^−4^	1.85 × 10^−4^	1.06 × 10^−4^	2.10 × 10^−4^
		Children	9.72 × 10^−4^	1.21 × 10^−3^	6.72 × 10^−4^	1.38 × 10^−3^

## Data Availability

Dataset available on request from the authors.

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
