# Peer review of "Health Risk Assessment of Heavy Metals in Agricultural Soils Based on Multi-Receptor Modeling Combined with Monte Carlo Simulation"

_toxics, 2024, doi:10.3390/toxics12090643_

Round 1
Reviewer 1 Report
Comments and Suggestions for Authors
Review report manuscript ID: toxics-3151741
The findings could be interesting for researchers. However, the following comments should be addressed before proceeding with this manuscript for further. Authors are highly recommended to correct manuscript as per following suggestions for enhancing readability and reproducibility of results.
- Abbreviations used throughput the manuscript needs to be defined the first time they are used in the abstract and/or other sections.
- Keywords should be arranged in alphabetical order.
- Please ensure that the meanings and reference values for all parameters in the equation are included in the Materials and Methods section of the main manuscript.
-Please provide the brands of chemicals and equipment’s used during the experiments throughout the materials and methods section.
- The following references may be helpful for this paper and recommended to be cited.
https://doi.org/10.1016/j.jclepro.2023.139512; https://doi.org/10.1016/j.ecoenv.2023.115228
- The authors have presented the results, but I did not find any discussion regarding them. I strongly recommend that the authors compare their findings with those of previous studies.
Author Response
Comments 1: Abbreviations used throughput the manuscript needs to be defined the first time they are used in the abstract and/or other sections.
Response 1: Thank you for the comments. Abbreviations used throughput the manuscript have been defined the first time they are used in the abstract and/or other sections. as shown in Lines 16-20. The details are as follows:
The principal component analysis-absolute principal component score-multiple linear regression (APCS-MLR) and the positive matrix factorization (PMF) revealed four sources of heavy metals: agricultural pollution sources Cadmium, (Cd), industrial and mining pollution sources Arsenic, (As), transportation pollution sources Zinc and Lead (Zn and Pb), and natural pollution sources Chromium, Nickel and Copper (Cr, Ni, and Cu).
Comments 2: Keywords should be arranged in alphabetical order.
Response 2: Thank you for the comments. The keywords of the manuscript have been arranged in alphabetical order. The details are as follows:
Keywords: Farmland soil; Heavy metals; Pollution sources; Risk assessment; Spatial characteristics
Comments 3: Please ensure that the meanings and reference values for all parameters in the equation are included in the Materials and Methods section of the main manuscript.
Response 3: Thank you for the comments. It has been ensured that the meanings and reference values of all parameters in all equations are included in the Materials and Methods section of the manuscript. The details are as follows:
As shown in Line 153. The formula is PI=Ci/Si, and the specific parameters and meanings have been added to Lines 153-159 of the manuscript. The specific parameters and meanings of the rest of the equations have been added to the Materials and Methods section of the manuscript.
Comments 4: Please provide the brands of chemicals and equipment’s used during the experiments throughout the materials and methods section.
Response4: Thank you for the comments. Brands of chemicals and equipment used in the experiments have been provided in the “Materials and Methods” section. As shown in Lines 131-144. The details are as follows:
All metals were determined after sample pretreatment. Cd samples were treated with HF, HNO3, and HClO4, As samples with aqua regia, and Cr, Ni, Cu, Zn, and Pb samples with powder pressure method. Inductively coupled plasma mass spectrometry (ICP-MS, Agilent Technologies 7700 Series, USA) was utilized to quantifie the concentration of Cd, atomic fluorescence spectrometry (AFS-8220 Beijing Titan Instruments, China) to the concentration of As, and X-ray fluorescence spectrometry (HS-XRF, MERAK-SC Beijing Anke Huisheng Technology, China) to the concentration of Cr, Ni, Cu, Zn, and Pb. Quality control was performed by inserting one standard reference material and one duplicate sample into each 50 samples to calculate the passing rate of accuracy, precision and relative deviation. The passing rate of accuracy and precision was calculated to be 100%, and the passing rate of relative deviation was 95%, which was in accordance with the requirements of the soil standard DZ/T 0295–2016. The method detection limits (MDL) were 2.00, 2.00, 2.00, 2.00, 0.05, 2.00, and 0.003 mg/kg for Cr, Ni, Cu, Zn, As, Pb, and Cd, respectively.
Comments 5: The following references may be helpful for this paper and recommended to be cited.
https://doi.org/10.1016/j.jclepro.2023.139512; https://doi.org/10.1016/j.ecoenv.2023.115228
Response 5: Thank you for the comments. The following references were helpful for our manuscript, especially the one cited below. As shown in Lines 33-36.
https://doi.org/10.1016/j.jclepro.2023.139512.
However, the second reference below was not very helpful to our manuscript and therefore not cited.
https://doi.org/10.1016/j.ecoenv.2023.115228
Comments 6: The authors have presented the results, but I did not find any discussion regarding them. I strongly recommend that the authors compare their findings with those of previous studies.
Response 6: Thank you for the comments. We compared and discussed the findings in the manuscript with previous studies, as shown in Line 304. Specific related discussions have been added to the Results and Analysis section of the manuscript.

Reviewer 2 Report
Comments and Suggestions for Authors
Once I have reviewed the paper, which I find interesting to know about the presence of heavy metals in agricultural soils, I am going to make some comments that I think can improve the paper.
Regarding format:
Very large font size in some tables such as table 1, table 2 and table 3.
The bibliography does not adapt to the format, the name of the journal must be abbreviated, the year in bold, for example. Review it. I think there is also an excess of references from Chinese authors and the references should be more international in nature.
Line 280 add the unit mg kg-1ª heavy content in the study area.
Section 3 should be called “Results and discussion” instead of “Results”, since the results are compared with other authors, although not too much, perhaps a little discussion is missing.
The number of samples seems very appropriate to me and the presentation of the results is very visual and the statistics are adequate.
What types of soils are studied according to FAO or USDA classifications? It is not enough to say that they are brick red soil, reddish soil, brick yellow soil, etc.
There is talk of agricultural soils, but what crops are they dedicated to? What agricultural practices do they have? Or what water are they irrigated with? They should be described more.
What texture do the soils have? It is important to know it in order to retain heavy metals.
It is very interesting to apply geoenvironmental indices and health indices. What background has been used in these indices, the soils of Hainan or China?
Author Response
Comments 1: Very large font size in some tables such as table 1, table 2 and table 3.
Response 1: Thank you for the comments. We have adjusted the font size of tables 1, 2, 3, 4, 5, 6, 7, and 8 in the manuscript.
Comments 2: The bibliography does not adapt to the format, the name of the journal must be abbreviated, the year in bold, for example. Review it. I think there is also an excess of references from Chinese authors and the references should be more international in nature.
Response 2: Thank you for the comments. The formatting of references in the manuscript has been adjusted, e.g., journal titles must be abbreviated and years bolded. Some of the references to Chinese authors have also been replaced with more international references.
Comments 3: Line 280 add the unit mg kg-1ª heavy content in the study area.
Response 3: Thank you for your comments. We have added the units mg kg-1 for the heavy metal content in the study area in Line 280 in the original manuscript as shown in Line 334 in the manuscript.
Comments 4: Section 3 should be called “Results and discussion” instead of “Results”, since the results are compared with other authors, although not too much, perhaps a little discussion is missing.
Response 4: Thank you for the comments.Discussion has been supplimented in Section 3.
Comments 5: The number of samples seems very appropriate to me and the presentation of the results is very visual and the statistics are adequate. What types of soils are studied according to FAO or USDA classifications? It is not enough to say that they are brick red soil, reddish soil, brick yellow soil, etc.
Response 5: Thank you for the comments. Based on FAO or USDA classifications, we studied brick red soil, red soil, brick yellow soil, coastal sandy soil, coastal saline soil, black stone ash soil, volcanic ash soil, and rice soil in our study area. As shown in Lines 115-118. The details are as follows:
The main soil types are brick red soil, red soil, brick yellow soil, coastal sandy soil, coastal salt soil, black lime soil, volcanic ash soil and rice soil. The plains are interspersed with hills and the terrain is complex, mainly characterized by high in the middle and low around [30].
Comments 6: There is talk of agricultural soils, but what crops are they dedicated to? What agricultural practices do they have? Or what water are they irrigated with? They should be described more.
Response 6: Thank you for the comments. Agricultural soils in the study area, who specialize in rice, winter fruits and vegetables, fruit trees, etc., and agricultural practices are mainly crop rotation and intercropping, relying mainly on water resources such as reservoirs, natural rivers, and rainfall for irrigation. As shown in Lines 108-114. The details are as follows:
The production of crops, mainly rice, winter melons and vegetables, and fruits is very well developed and therefore also includes the extensive use of pesticides, fertilizers and other agrochemicals. Agricultural production methods are mainly crop rotation and intercropping, and irrigation relies mainly on water resources such as reservoirs, natural rivers and rainfall. In addition, industrial activities such as metal mining, related smelters, paper mills, cement factories, etc. contribute significantly to the development of the local economy.
Comments 7: What texture do the soils have? It is important to know it in order to retain heavy metals.
Response 7: Thank you for the comments. This research focuses on the actual pollution status of heavy metals and the fact of health hazards caused by heavy metals. According to the suggestions of the reviewers, we will strengthen the acquisition and supplement of natural information such as soil texture and related data information such as human activities in the future, and build a more comprehensive variable database to provide reference for the interpretation of risk results, which is a very important part of our future research.
Comments 8: It is very interesting to apply geoenvironmental indices and health indices. What background has been used in these indices, the soils of Hainan or China?
Response 8: Thank you for the comments. The Earth Accumulation Index (Igeo) uses background values for soils on Hainan Island, while the Health Risk Index uses internationally specified background values.

Reviewer 3 Report
Comments and Suggestions for Authors
The article contains interesting and comprehensive research. However, the article should be supplemented with a discussion and reference to world studies. For the purpose of the authors write: "The results of the study can provide theoretical and practical foundations important for identifying sources of soil contamination with heavy metals, assessing risks and implementing comprehensive management strategies in similar rapidly developing economic regions" should describe the answer to this purpose in the discussion of the article.
Author Response
Comments 1: However, the article should be supplemented with a discussion and reference to world studies.
Response 1: Thank you for the comments. Our manuscript has been supplemented with discussions and references to world studies. Details were shown in section 3 “Results and discussion” of the manuscript.
Comments 2: For the purpose of the authors write: "The results of the study can provide theoretical and practical foundations important for identifying sources of soil contamination with heavy metals, assessing risks and implementing comprehensive management strategies in similar rapidly developing economic regions" should describe the answer to this purpose in the discussion of the article.
Response 2: Thank you for the comments. "The results of the study can provide theoretical and practical foundations important for identifying sources of soil contamination with heavy metals, assessing risks and implementing comprehensive management strategies in similar rapidly developing economic regions" has been described in section 3 “Results and Discussion” as shown in Lines 304-574 of the manuscript.
Round 2
Reviewer 1 Report
Comments and Suggestions for Authors
Dear editor,
Thanks for inviting me to re-evaluate this paper. It is ready to be accepted.
Reviewer 3 Report
Comments and Suggestions for Authors
I accept this version